# Uncovering near-free platinum single-atom dynamics during electrochemical hydrogen evolution reaction

Shi Fang [1], Xiaorong Zhu[2], Xiaokang Liu[1], Jian Gu[3], Wei Liu [1], Danhao Wang[1], Wei Zhang[1], Yue Lin [3], Junling Lu [3], Shiqiang Wei [1], Yafei Li[2] & Tao Yao [1✉]

Single-atom catalysts offering intriguing activity and selectivity are subject of intense investigation. Understanding the nature of single-atom active site and its dynamics under working state are crucial to improving their catalytic performances. Here, we identify at atomic level a general evolution of single atom into a near-free state under electrocatalytic hydrogen evolution condition, via operando synchrotron X-ray absorption spectroscopy. We uncover that the single Pt atom tends to dynamically release from the nitrogen-carbon substrate, with the geometric structure less coordinated to support and electronic property closer to zero valence, during the reaction. Theoretical simulations support that the Pt sites with weakened Pt–support interaction and more $5d$ density are the real active centers. The single-atom Pt catalyst exhibits very high hydrogen evolution activity with only 19 mV overpotential in 0.5 M $H_2SO_4$ and 46 mV in 1.0 M NaOH at 10 mA cm$^{-2}$, and long-term durability in wide-pH electrolytes.

[1] National Synchrotron Radiation Laboratory, University of Science and Technology of China, Hefei 230029, P.R. China. [2] Jiangsu Collaborative Innovation Centre of Biomedical Functional Materials, Jiangsu Key Laboratory of New Power Batteries, School of Chemistry and Materials Science, Nanjing Normal University, Nanjing 210023, P.R. China. [3] Department of Chemical Physics, University of Science and Technology of China, Hefei 230026, P.R. China. ✉email: yaot@ustc.edu.cn

The advances in the rational design and controllable synthesis of catalysts at the atomic level have led to a booming development in the fundamental and applied researches of catalysis[1–5]. Recently, single-atom catalysts (SACs) that comprise isolated active metal centers have attracted considerable research interests, owing to its maximum atom efficiency and unique atomic structures and electronic properties[6–9]. For instance, downsizing the expensive Platinum (Pt) nanoparticle to single-atom dispersion provides an effective way to not only create cost-effective catalysts, but also improve its activity and selectivity toward several electrochemical reduction processes, such as hydrogen evolution reaction (HER), oxygen reduction reaction, nitrogen reduction reaction, and $CO_2$ reduction reaction[10–13], etc. Despite these progress, further exploration of novel SACs is still an important frontier in catalysis research, both from the aspects of materials synthesis and mechanistic understanding. To this end, an atomic-level identification of the nature of the single-atom active site is imperative so as to aid the targeted design of SACs.

For the SACs reported so far, the atomically dispersed metal atoms are usually anchored on specific substrates by chemical bonding with neighboring atoms, such as the metal-nitrogen-carbon (M-N-C) materials with the trace metal and nitrogen co-doped on the carbon support[14,15]. The catalytic activity and selectivity of SACs are highly dependent on the local coordination environment of the metal centers, in other words, the electronic and geometric interactions between the single atoms and support[16]. As such, the metal–support interaction by electron transfer between single metal atom and the nearest coordination atoms will affect the charge density and distribution of metal sites, which will further impact the overall catalytic properties of the SACs[16,17]. The strong interaction between the single metal atom and supported N/C atoms can lead to a large charge transfer from metal to N/C atoms. Consequently, the metal atom is positively charged, presenting cation-like high-valence states. Although these high-valence metal atoms are beneficial for some electro-oxidation reactions[4,18,19], they might be unfavorable for electro-reduction reactions, owing to the less $d$ electrons participating in the reaction. If the metal–support interaction becomes weak, the electronic structure of the center metal atom might resemble that of a free atom, making bonding with reactants more energetically preferred in electro-reduction reactions[20]. However, such free single atoms are rarely reported in the as-prepared SACs previously, mostly from the consideration of stability issue. Indeed, it has been reported that single-atom active site would undergo reconstruction in atomic configuration, accompanied by the variations in valence state, under the working conditions[1,8,21]. This means that the interaction between single metal atom and support would be self-adjusted in order to boost the catalytic reaction. Therefore, a deeper understanding of the working state of SACs may provide some new insights into the SACs.

Here, we report the discovery of a near-free single-atom Pt as a highly efficient active center under the electrochemical HER condition, by using operando X-ray absorption fine structure (XAFS) technique. The operando XAFS allows us to follow the structural evolutions of single-site metal centers, and unravels that the single-atom Pt becomes low-coordinated with its surrounding atoms, together with the close-to-zero valence state. As a result, the interaction between single metal atom and N-C support is weakened, leading to a near-free state of Pt, which energetically facilitates $H_2O$ adsorption in alkaline electrolyte, followed by a more optimized H adsorption. As such, single-atom Pt anchoring on the specific site of N-C framework, accessed via the atomic layer deposition (ALD) method, exhibits the superior HER performance with ultra-low overpotentials and long-term stability in wide-pH-range electrolytes.

**Structural characterization of $Pt_1$/N-C single-atomic site catalyst**. We synthesized atomically dispersed Pt on a metal-organic framework (MOF) derived N-C framework (sample designated as $Pt_1$/N-C using ALD. It is noted that each carbon ring of Uio-66-$NH_2$ anchors one -$NH_2$ group, and the -$NH_2$-derived uncoordinated N site may act as the anchoring site for Pt atom during the ALD process (Supplementary Fig. 1). The combination of well-defined MOF structure and ALD technique helps atomic-level precise deposition of single-atom Pt on the specific N-C site with nearly uniform local coordination environments. The atomic-resolution scanning transmission electron microscopy (STEM) performed in high-angle annular dark field (HAADF) mode show that most Pt species are isolated and uniformly dispersed on the N-C substrate (bright spots corresponding to single-atom Pt) (Fig. 1a). The STEM-coupled energy-dispersive spectroscopy (EDS) element mapping further corroborates the presence of C, N, and Pt elements, and the homogeneous distribution of Pt throughout the octahedral-shaped sample (Fig. 1b). The single-Pt-atom site was further supported by CO-absorbed diffuse reflectance infrared Fourier transform spectroscopy (CO-DRIFTS). As shown in Supplementary Fig. 2a, the narrow and quasi-symmetrical band at $\sim 2084\,cm^{-1}$, which remain unchanged during CO desorption, can be ascribed to linearly adsorbed CO on single-atom Pt. For comparison, we synthesized Pt nanoparticles (Pt-NPs) by treating the $Pt_1$/N-C in a 20% $H_2$/Ar flow at 300 °C (Supplementary Fig. 3). The CO-DRIFTS spectra of Pt-NPs exhibit a redshift of the IR peak from 2058 to $2052\,cm^{-1}$ (Supplementary Fig. 2b) during CO desorption, which correlate with changes in dipole–dipole coupling between CO on the surface of Pt crystal[22–25]. Pt content in $Pt_1$/N-C is $\sim 2.5$ wt% as determined by the coupled plasma optical emission spectrometry (ICP–OES) analysis. Note that when using the Uio-66 derived C framework as the substrate, small Pt clusters are formed (Supplementary Fig. 4) under the same ALD process, confirming that the uncoordinated N atoms in the N-C framework provide essential sites for immobilizing single-atom Pt[3,26,27]. Furthermore, Pt-N coordination can be confirmed by N K-edge X-ray absorption spectroscopy and N1$s$ X-ray photoelectron spectroscopy results (Fig. 1c and Supplementary Fig. 5). The N K-edge absorption spectrum of pure N-C framework without Pt can be characterized by four nitrogen features: pyridinic (peak a, 398.5 eV), pyrrolic (peak b, 399.4 eV), graphitic (peak c, 401.6 eV), and C-N-C or C-N σ* bond (peak d, 407.3 eV), respectively[28–30]. After Pt deposition, the pyrrolic peak b remains constant, whereas the pyridinic peak a is damped and split into two peaks (a$_1$ and a$_2$), where a$_2$ is derived from a portion of pyridinic N bonded to Pt atoms, in accordance with the previous reports[31,32].

**Operando X-ray absorption spectroscopy study**. To reveal the nature of the single-Pt-atom site under electro-reduction working conditions, the operando XAFS measurements using a purpose-built cell were performed on $Pt_1$/N-C catalyst during HER. Carbon cloth was used as the working electrode for loading catalyst, together with Ag/AgCl reference electrode and carbon rod counter electrode in an electrochemical cell (Supplementary Fig. 6). As shown in Supplementary Fig. 7, the catalysts were homogeneously adhered in the gap of the carbon fibers without obvious aggregation, meaning that the catalysts may be fully exposed to the electrolyte so that subtle variations of the Pt active sites can be discerned.

Figure 1d shows the evolutions of normalized Pt L$_3$-edge X-ray absorption near-edge structure (XANES) spectra with applied potential. The XANES spectrum of catalyst is characterized by the strong white-line peak, which corresponds to transition from the

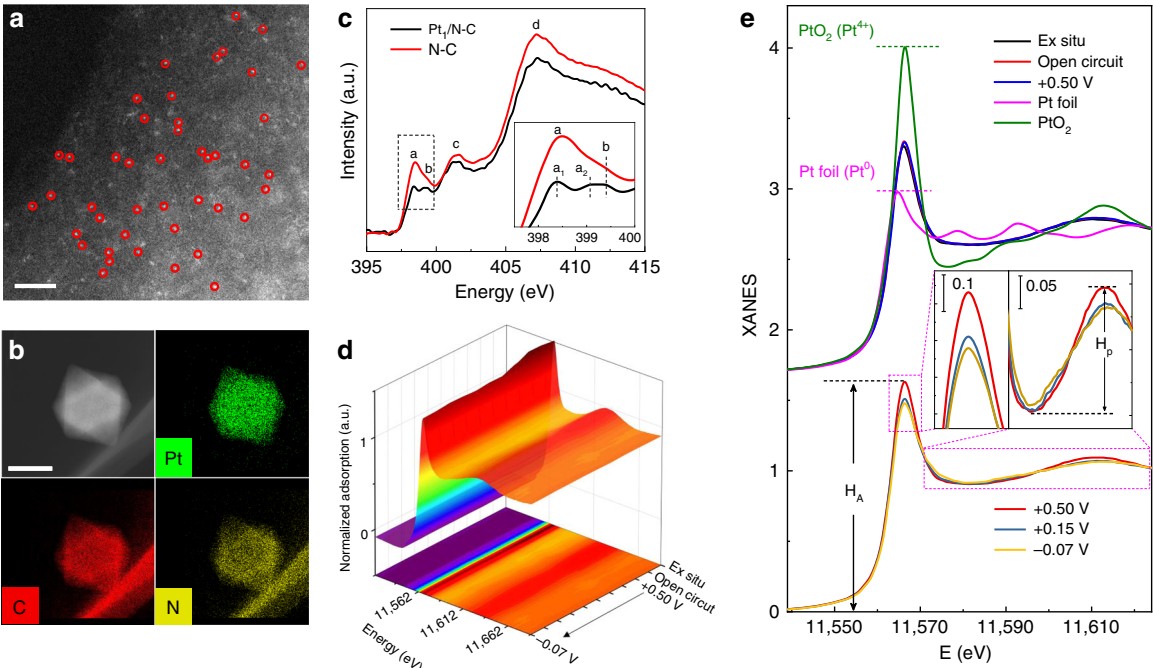

**Fig. 1 Microstructure and operando XANES characterization. a** Atomic-resolution HAADF–STEM image of Pt$_1$/N-C, showing that only Pt single atoms are present (marked by red circles). Length of scale bar is 2 nm. **b** Micromorphology and elemental mapping of Pt$_1$/N-C. Length of scale bar is 200 nm. **c** N K-edge XANES spectra of N-C framework (before ALD) and Pt$_1$/N-C (after ALD). **d** Three-dimensional profile plot of successive operando Pt L$_3$-edge XANES spectra acquired in different conditions. **e** Selected XANES spectra in **d** at different applied voltages from the open-circuit condition to −0.07 V during HER, and the XANES data of the reference standards of Pt foil and PtO$_2$. Inset, Magnified white-line peak and post-edge XANES region.

occupied Pt $2p_{3/2}$ core–electron to empty $5d$ states, and thus is indicative of $5d$-band occupancy[33,34]. It can be obviously discerned that the overall white-line intensity (H$_A$) decreases as applying negative potential during the reaction (Fig. 1d). Specifically, the white-line intensities of the catalyst under ex situ, open circuit and +0.5 V (vs. RHE, all voltages mentioned hereafter are normalized to RHE scale) conditions remain unchanged (Fig. 1e), suggesting that the single-atom Pt site is structurally stable without chemical adsorption of reactant species in the electrolyte. This is reasonable as +0.5 V is commonly considered as located in the double layer region where only capacitive charging process occurs on the electrode surface without any redox reaction[35,36]. When the applied potential negatively shifts to +0.15 V and to −0.07 V, the white-line intensity gradually decreases, indicating a higher $5d$ occupancy of Pt. Here, the unoccupied $5d$ states mainly arise from the hybridization between Pt and nearby C/N atoms from support. Hence, higher $5d$ occupancy implies the less charge transfer from the single-atom Pt to the nearby C/N atoms at +0.15 V and −0.07 V.

To obtain the quantitative information of electronic structural evolutions, we examine the valence state and formal $d$-band holes count of Pt. The above-described variations can be described more clearly from the differential XANES (ΔXANES) spectra by subtracting the spectra collected at different potentials from that of Pt foil reference (Supplementary Fig. 8). The valence states and formal $d$-band hole counts of Pt can be determined quantitatively by integrating the area of the white-line peak in the ΔXANES spectra. It can be found that the mean valence state of Pt decreases from 1.89 to 1.17 and 1.12 for the catalyst at the potentials from +0.5 V to +0.15 V and −0.07 V (Fig. 2b), respectively, implying that the Pt site is reduced close-to-zero valence toward the metallic state. The formal $d$-band hole count was estimated using a slope of 1.174 unit area per $d$-band hole obtained from Pt$^0$ foil ($5d^96s^1$) and Pt$^{IV}$O$_2$ ($5d^66s^0$) standards. The absolute number of $d$-band holes for the ex situ Pt$_1$/N-C is 2.420, which is reduced to 1.877 and 1.838 for the Pt$_1$/N-C under

+0.15 V and −0.07 V (Fig. 2a), suggesting the increased $d$-band electrons of Pt, verifying the weaker interaction between the Pt and the N-C substrate under working states.

The evolution of coordination configuration of single-atom Pt was further identified by extended X-ray absorption fine structure (EXAFS) (Supplementary Fig. 9 and 10). The Fourier-transformed (FT) $k^2$-weighted EXAFS spectra for Pt$_1$/N-C at the corresponding applied potentials all present a single dominant peak at ca. 1.6 Å that can be assigned to the coordination between Pt atom and C/N/O light atoms. However, the intensity of FT peak damps by ~13 and 35% when the applied potentials negatively shift from +0.5 V to +0.15 V and to −0.07 V (Fig. 2c), respectively, suggesting the distinct variations of the local geometric structure of Pt site under working conditions. The geometric variations can also be reflected by the increased structural disorder, as revealed from the features in the XANES post-edge plateau corresponding to the interference of photo-electrons with local atoms. As shown in Fig. 1e inset, the intensity of the oscillation hump (H$_P$) decreases with the increased potential, confirming dynamically disordered structures around the Pt atom[33]. Note that the above evolutions of the atomic and electronic structures are reversible as demonstrated in Supplementary Fig. 11 and 12.

Then, we would like to reveal the detail evolutions of geometric configuration of single-Pt site. To first confirm the geometric structure of the ex situ state, we performed EXAFS curve fitting considering two backscattering paths of Pt–C and Pt–N from N-C framework, for the Pt$_1$/N-C under ex situ, open-circuit, and +0.5 V conditions (Fig. 2c and Supplementary Fig. 13). The best fitting analyses give four coordination number of Pt-C/N (Supplementary Table 1). Considering the limit of XAFS in differentiating Pt-C and Pt-N coordination because of the close scattering amplitude of C and N, we resort to the density functional theory (DFT) calculations considering several configurations, including Pt$_1$-C$_3$N$_1$, Pt$_1$-C$_2$N$_2$, and Pt$_1$-C$_1$N$_3$

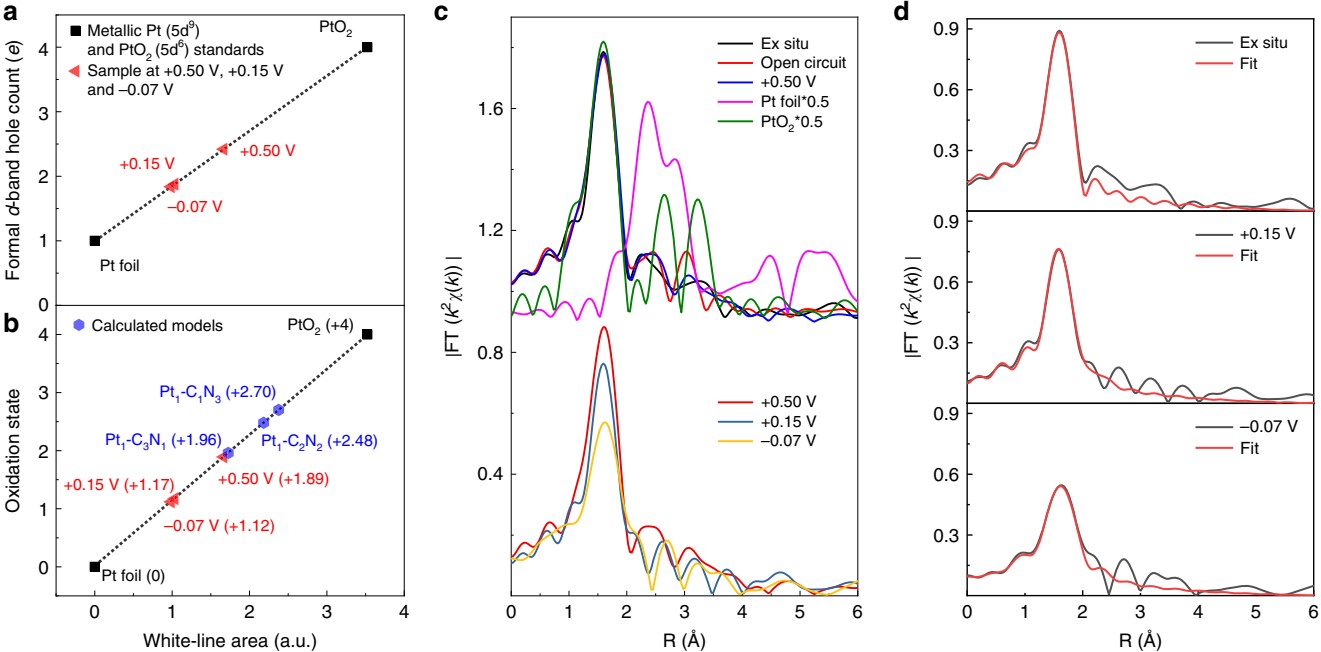

**Fig. 2 Operando EXAFS and ΔXANES characterization. a** The fitted average formal $d$-band hole counts and **b** oxidation states of Pt from ΔXANES spectra, depicted with the calculated oxidation states of Pt$_1$-C$_3$N$_1$, Pt$_1$-C$_2$N$_2$, and Pt$_1$-C$_1$N$_3$ models. The oxidation state of Pt$_1$-C$_3$N$_1$ are the most proximate to that of the ex situ sample. **c** Corresponding $k^2$-weighted Fourier transform (FT) spectra of Fig. 1e. **d** First-shell fitting of EXAFS spectra under ex situ, +0.15 V and −0.07 V conditions.

(Supplementary Fig. 14). Quantitatively, the Pt atom donates 0.47, 0.59 and 0.64 $e$ to the support in Pt$_1$-C$_3$N$_1$ Pt$_1$-C$_2$N$_2$, and Pt$_1$-C$_1$N$_3$, respectively (where $e$ is the elementary charge) (Supplementary Table 2). The corresponding oxidation states of Pt estimated by normalizing Bader charges (see Supplementary Note 2) are 1.96 to 2.48 and 2.70 for Pt$_1$-C$_3$N$_1$, Pt$_1$-C$_2$N$_2$, and Pt$_1$-C$_1$N$_3$, respectively (Fig. 2b and Supplementary Note 2). Hence, the calculated Pt valence state of Pt$_1$-C$_3$N$_1$ (1.96) is more close to that (1.89) derived from XANES analyses. Based on these considerations, the best EXAFS fitting gives three Pt-C bonds ($R = 2.04$ Å) and one Pt-N bond ($R = 2.09$ Å) (Supplementary Table 1). Moreover, structures containing more than one Pt-N coordination, i.e., Pt$_1$-C$_2$N$_2$, and Pt$_1$-C$_1$N$_3$ can be reasonably excluded since it is unlikely for one Pt bridging two N atoms with N-N distance of ca. 10 Å between two adjacent N atoms in Uio-66-NH$_2$.

Interestingly, for the catalyst under +0.15 V, the Pt–C/N coordination number reduced to 2, together with the appearance of one additional Pt–O coordination ($R = 2.06$ Å) (Fig. 2d and Supplementary Fig. 15), which is possibly arisen from the adsorption of the oxygen-related group from H$_2$O or OH$^-$. Considering that OH$^-$ adsorption should occur in a more positive potential region of 0.65 ∼ 0.85 V[35], and in alkaline HER, the catalytic cycle is initiated by adsorption of H$_2$O onto the active site. Hence, we can conclude that the Pt–O coordination can be ascribed to adsorption of H$_2$O molecules. This is also consistent with the rate limitation step of dissociation of the absorbed H$_2$O at this stage. The reduced coordination number of Pt with support indicate that Pt atoms are released from support, this effect we consider is beneficial for the adsorption of H$_2$O molecules. Furthermore, only twofold Pt–C/N coordination was obtained for the catalyst at −0.07 V, where the water is dissociated into OH$^-$ and adsorbed H*, followed by the attack of another proton to generate H$_2$. However, Pt–H coordination can hardly be discerned, owing to the small scattering amplitude of H[37]. Hence, only twofold Pt–C/N coordination is considered in

the EXAFS fitting (Fig. 2d and Supplementary Fig. 16). These coordinative evolutions of Pt can be further confirmed by the wavelet transform analysis, as shown in Supplementary Fig. 17.

We want to note that these findings of near-free Pt state during electro-reduction reaction hold true in wide-pH environments. As shown in Supplementary Fig. 18, the operando XAFS measurements were also carried out in 0.1 M HClO$_4$ (pH = 1) and 1.0 M NaOH (pH = 14) under the applied potential of +0.15 V. The variation trends of XANES and EXAFS spectra in pH1 and pH14 are analogous to those in pH 13, showing the significantly reduced intensities of white-line peak and FT peak. Through the quantitative analyses, we demonstrate that the single-atom Pt resemble the free atom during HER, with low coordination to the support and the valence state close to zero. We reckon that such evolutions of the single-Pt sites might directly boost the HER in wide-pH electrolytes.

**Theoretical simulations.** Theoretical investigations based on DFT calculations were carried out to unravel the influences of the evolved configurations on the electronic structure. According to the above EXAFS fitting models, two possible and stable two-coordinate configurations, i.e., Pt$_1$-C$_2$ and Pt$_1$-C$_1$N$_1$, were constructed for simulating the varied Pt local structure under realistic reaction, as shown in Fig. 3a. On the basis of Bader charge analysis, the charge transfer from the Pt atom to support is only ca. 0.1 ∼ 0.2 $e$ for the Pt$_1$-C$_2$ and Pt$_1$-C$_1$N$_1$, much smaller than 0.47 $e$ for Pt$_1$-C$_3$N$_1$ (Supplementary Table 2). As such, the Pt–support interactions of the two-coordinate models are much weaker than that of the ex situ Pt$_1$-C$_3$N$_1$ model.

Based on the models in Fig. 3a, we then explore the potential HER activity of less-coordinate Pt$_1$-C$_2$ and Pt$_1$-C$_1$N$_1$ sites. The overall HER pathway can be described by a three-state diagram comprising the initial catalyst-water state, to the catalyst-H intermediate state, and the final catalyst+H$_2$ state. The Gibbs free-energy of atomic hydrogen adsorption, $|\Delta G_{H^*}|$, has been

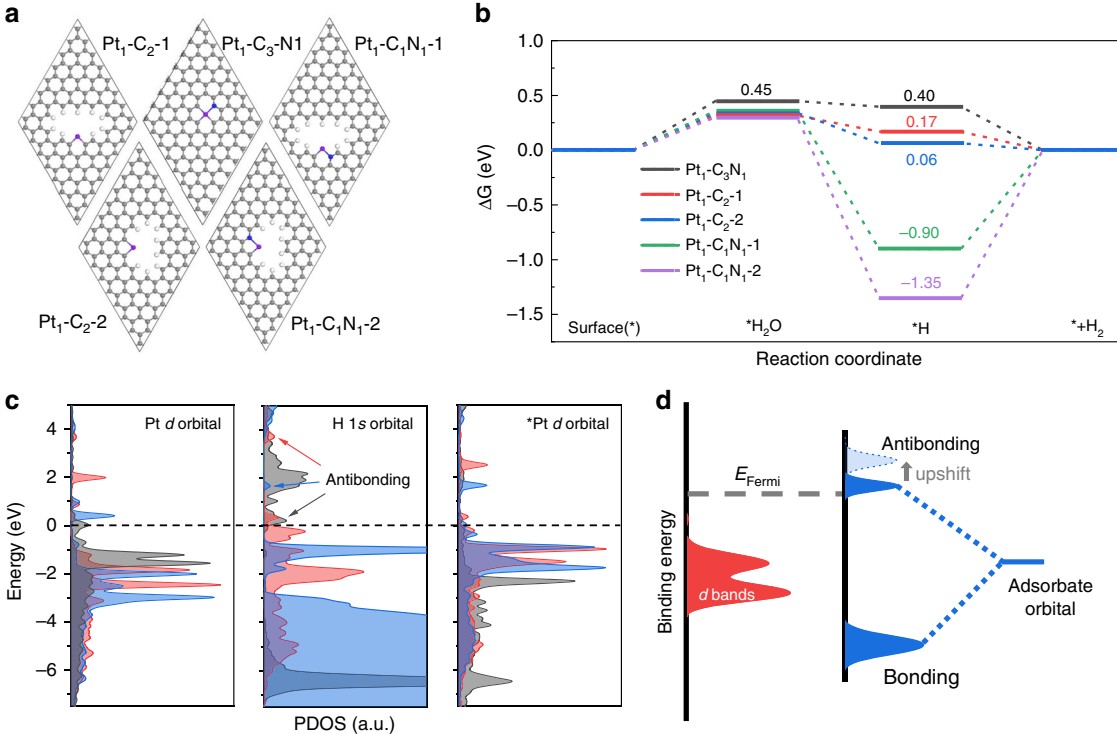

**Fig. 3 Theoretical investigations. a** Computational models of the $Pt_1$-$C_3N_1$ and two $Pt_1$-$C_2$ and two $Pt_1$-$C_1N_1$ moieties. H, white; C, gray; N, blue; Pt, purple. **b** Calculated adsorption energies of $H_2O$ and H on the surface of $Pt_1$-$C_3N_1$, $Pt_1$-$C_2$ and $Pt_1$-$C_1N_1$. **c** Calculated PDOS of Pt $d$ orbital of clean surfaces, H $1s$ of hydrogen adsorbed on Pt sites and Pt $d$ of Pt sites with adsorbed hydrogen (denoted *Pt $d$ orbital), from left to right, respectively. The gray, red and blue contour represent $Pt_1$-$C_3N_1$, $Pt_1$-$C_2$-1, and $Pt_1$-$C_2$-2 surfaces, respectively. The DOS peaks of H1s-Pt 5d antibonding are indicated by the arrows. Note that the sharp peak ~ −1 eV for $Pt_1$-$C_2$-2 represent bonding rather than antibonding state, as it corresponds to the prominent Pt $d$ band in the *Pt $d$ PDOS. **d** Schematic DOS illustration of the interaction between Pt and H, the H $1s$ states split into bonding and antibonding states. With four-coordinate Pt evolving to two-coordinate, the corresponding antibonding states upshift to a higher energy with a lower occupancy.

considered as a reasonable descriptor of the HER activity. As is known, Pt shows the optimal level for H adsorption step with a near-zero $\Delta G_{H^*}$ (ca. 0.09 eV), owing to its suitable $d$-band position[38,39]. As shown in Fig. 3b, the $\Delta G_{H^*}$ of the two $Pt_1$-$C_2$ sites (0.17 and 0.06 eV) are closer to 0 eV, compared to that of ex situ $Pt_1$-$C_3N_1$ site (0.40 eV). Moreover, both $Pt_1$-$C_1N_1$ sites exhibit significantly larger $|\Delta G_{H^*}|$, meaning that hydrogen adsorption on $Pt_1$-$C_1N_1$ sites are too strong to desorb. The DFT analyses, in combination with the above XANES, indicate that the Pt in $Pt_1$-$C_2$ sites have more 5$d$ densities close to free-atom Pt, leading to a more favorable H* adsorption, thus facilitating the HER in acid condition. For the HER in an alkaline media, the water dissociation kinetics from the Volmer step may determine the overall reaction rate. The DFT calculations show that the water adsorption on the two-coordinate sites is more favorable than that on the $Pt_1$-$C_3N_1$ site, as the $H_2O$ adsorption energy decrease from 0.45 eV to 0.3 ∼ 0.35 eV (Fig. 3b).

Above analysis on adsorption energy demonstrates that hydrogen adsorption on $Pt_1$-$C_2$ sites is significantly improved comparing with that on $Pt_1$-$C_3N_1$ site. To investigate the underlying reason, we carried out partial density of state (PDOS) analysis (Supplementary Fig. 19 and Fig. 3c). The dominant $d$ band features (−1 ∼ −3.5 eV) of the three surfaces are nearly degenerate, compared to the broad $d$ band of bulk Pt (Supplementary Fig. 19). The $d$ band positions of both $Pt_1$-$C_2$ are lower than that of $Pt_1$-$C_3N_1$, indicating weaker Pt–support interaction. Besides, an obvious peak appears near the Fermi level (0 eV) for the $Pt_1$-$C_3N_1$ surface (Supplementary Fig. 19, pointed by arrows), suggesting relatively strong hybridization between Pt and C/N orbitals and reflecting strong bindings between Pt atoms and

substrates. In contrast, the Pt sites of both two $Pt_1$-$C_2$ structures provide much smaller DOS at the Fermi level, further verifying the weakened Pt–support interaction, meaning that Pt atoms become freer from the supports under reaction conditions. We consider that such near-free Pt with more 5$d$ density and low-valence state is favorable for the adsorption of $H_2O$ and H reactants. This can be confirmed by the DOS of the $Pt_1$-$C_3N_1$ and $Pt_1$-$C_2$ surfaces (Supplementary Fig. 20). Compared with $Pt_1$-$C_3N_1$, the prominent peaks of $Pt_1$-$C_2$ surfaces at the Fermi level imply higher localized electron density on Pt site, which may benefit the activation of $H_2O$ molecules and thus boosts HER activity.

Moreover, we consider the cases of a hydrogen atom adsorbed on the $Pt_1$-$C_3N_1$ and two $Pt_1$-$C_2$ surfaces, and calculate the corresponding DOS projected onto the H $1s$ states and Pt $d$ bands (Fig. 3c, the middle and right panels). In the H $1s$ PDOS, the dominant features are the H $1s$-Pt 5$d$ bonding resonances below the Fermi level, distinct in energy position and intensity for three different surfaces. In particular, the antibonding states are all above the Fermi level but in different energy position (noted by the arrows), by which the occupancy of antibonding states can be discerned. For instance, the antibonding states of $Pt_1$-$C_3N_1$ is close to Fermi level, meaning a relative high occupancy. Although both $Pt_1$-$C_2$-1 and $Pt_1$-$C_2$-2 show higher energy position of antibonding, indicating lower antibonding occupancy and less repulsive interaction between H $1s$ and Pt 5$d$ orbitals, leading to the stronger Pt-H interaction. Note that the antibonding position of $Pt_1$-$C_2$-2 is lower than that of $Pt_1$-$C_2$-1, but the bonding state intensity of $Pt_1$-$C_2$-2 is much higher than that of $Pt_1$-$C_2$-1, resulting in a stronger hydrogen adsorption on $Pt_1$-$C_2$-2, consistent with the $\Delta G_{H^*}$ values.

Figure 3d shows schematic illustration of the interaction between Pt 5$d$ and H 1$s$ orbitals. When the hydrogen adsorbate hybridizes with the $d$ band of Pt, the adsorbate state split into localized bonding and antibonding states. The higher position of the antibonding, the stronger Pt-H interaction. Based on the above analysis, we reckon that with the evolution of Pt site from four-coordinate to two-coordinate, the antibonding states upshift to higher energy position, leading to lower occupancy and consequent stronger hydrogen adsorption. Hence, the theoretical insights emphasize the importance of the evolution of electronic and atomic structures of Pt into a near-free state under the electro-reduction potentials for the high HER activity.

**Electrochemical characterization.** Combining operando XAFS characterization and DFT calculation, we reveal for the first time that the single-atom Pt tend to evolve into a more freestanding state, which can substantially boost HER in both preferred water adsorption and moderate hydrogen adsorption. Thus, the electrocatalytic HER activities of $Pt_1$/N-C was examined in wide-pH electrolytes, including 1.0 M KOH and 0.5 M $H_2SO_4$ solutions. For comparison, the HER performance of the pristine N-C substrate and commercial 20% Pt/C were also measured under the same test conditions. As shown in Fig. 4a, at the current density of 10 mA cm$^{-2}$, the $Pt_1$/N-C delivers a very small overpotential of 46 and 19 mV in alkaline and acid solution, respectively, superior to those of Pt/C (57 and 25 mV). In contrast, the pristine N-C substrate exhibits negligible HER activity in both acid and alkaline, as the current densities of N-C are close to zero, confirming that the high HER activity of $Pt_1$/N-C is contributed by the single-atom Pt sites. Notably, the Tafel slope of $Pt_1$/N-C in 1.0 M KOH is 36.8 mV dec$^{-1}$, lower than that of Pt/C (39.8 mV dec$^{-1}$), indicating the Volmer–Heyrovsky mechanism as the HER pathway[39].

The Tafel slope of $Pt_1$/N-C (14.2 mV dec$^{-1}$) in 0.5 M $H_2SO_4$ is also comparable to that of Pt/C (18.6 mV dec$^{-1}$) (Fig. 4b). The result of HER catalyzing stability test is present in Fig. 4c, which shows negligible loss of current density in both acid and alkaline electrolytes in 20 hours, demonstrating excellent long-time durability of the $Pt_1$/N-C catalysts. Also, the single-atom structure and dispersion are preserved, as can be confirmed by the electron microscopic and XAFS measurements for the $Pt_1$/N-C catalyst after the long-time HER test (Supplementary Fig. 21). Moreover, the turnover frequencies (TOFs) of $Pt_1$/N-C are 22.07 and 1.89 $H_2$ s$^{-1}$ in acid and alkaline, 10 and 12 times higher than those of Pt/C, respectively (Fig. 4d). To the best of our knowledge, the HER performances in the wide-pH electrolytes of our $Pt_1$/N-C catalyst are almost the best among the SACs reported recently, as summarized in Supplementary Table 3. These results confirm that the $Pt_1$/N-C possesses the best intrinsic HER activity that is mainly contributed by the reconstructed single-Pt sites.

**Discussion**

On the basis of the operando XAFS, we identify the atomic and electronic structural evolutions of the single-atom Pt site under electrochemical HER. We uncover that the Pt atom tends to be a near-free state resembling a free metallic atom, that is, the interaction and charge transfer between Pt and N-C support is weakened. Theoretical simulations support that the near-free single-atom Pt possesses the favorable bonding energies with the reactants, responsible for the superior HER activity. Our study highlights the importance of creating more free single metal atoms while remaining stable for boosting the electrochemical reduction reactions. These findings also suggest a new insight into tailoring the coordination environment of single-atom active sites for developing high-performance single-atom electrocatalyst.

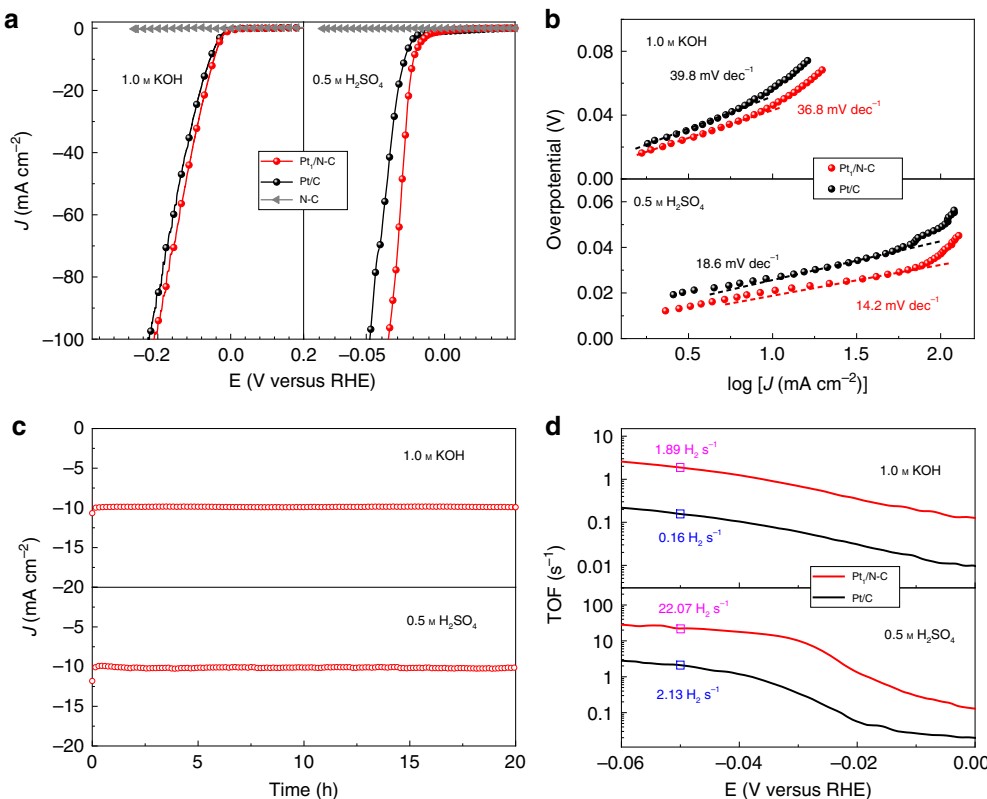

**Fig. 4 Electrochemical analysis in both 1.0 M KOH and 0.5 M H$_2$SO$_4$. a** LSV curves of the Pt$_1$/N-C, Pt/C and N-C framework. $J$, current density; $E$, potential. **b** Tafel plots for Pt$_1$/N-C and Pt/C electrocatalysts. **c** Durability test of Pt$_1$/N-C. **d** TOF plots of the Pt$_1$/N-C and Pt/C electrocatalysts.

## Methods

**Synthesis of Uio-66-NH$_2$ derived N-C framework**. At first, a mixture of ZrCl$_4$ (37.76 mg, 0.162 mmol) and H$_2$BDC-NH$_2$ (29.34 mg, 0.162 mmol) was dissolved in 36 mL DMF plus 4.0 mL HAc in a 60 mL glass vial, then sonicated for 1 h. The vial was sealed and heated at 120 °C for 24 h in an oven. The product was then centrifuged, washed with a mixture of methanol and DMF (volume ratio = 1:4), and then dried under vacuum at 90 °C. Then, powder of Uio-66-NH$_2$ was placed in a tube furnace and heated to 700 °C in a heating procedure of 2 °C/min and preserved for 3 h, under a flowing argon atmosphere. After cooling to room temperature, the black powder was collected and dispersed in 30 mL H$_2$O containing 0.2 mol HF for 8 h, and then centrifuged, washed with methanol and H$_2$O (volume ratio = 1:4) and dried under vacuum at 100 °C.

**ALD preparation of Pt$_1$/N-C**. Pt ALD was carried out on a viscous flow reactor (GEMSTAR-6 Benchtop ALD, Arradiance) by alternatively exposing to MeCpPtMe$_3$ precursor and O$_2$ (99.999%) at 150 °C. Ultrahigh purity N$_2$ (99.999%) was used as the carrier gas at a flow rate of 200 mL/min. The Pt precursor was heated to 65 °C to get a sufficient vapor pressure. The reactor inlets were held at 110 °C to avoid any precursor condensation. The timing sequence was 80, 120, 60, and 120 s for the MeCpPtMe$_3$ exposure, N$_2$ purge, O$_2$ exposure, and N$_2$ purge, respectively.

**Material characterizations**. The morphologies of the samples were examined by scanning electron microscopy and TEM on FEI-30 ESEM and JEOL-2100F systems at an accelerating voltage of 200 kV. EDS elemental mapping were obtained on a 26FEI Talos F200X device at 200 kV. HAADF–STEM, high-resolution TEM results were obtained on a JEM-ARM 200 F instrument at 200 kV. The concentration of Pt atoms was directly measured by ICP–OES (Optima 7300 DV, PerkinElmer). XPS measurements were carried out on an ESCALAB MKII instrument equipped with an Mg Kα source (hν = 1253.6 eV). The binding energy scale of all measurements was calibrated by referencing C 1 s to 284.5 eV. N K-edge XANES measurements were performed at the photoemission endstation at BL10B and BL12B beamlines of the National Synchrotron Radiation Laboratory (NSRL), China.

DRIFTS CO chemisorption measurements were performed on a Bruker IFS 66 v/s FTIR spectrometer equipped with a liquid nitrogen cooled HgCdTe (MCT) detector and a low-temperature reaction cell (Praying Mantis Harrick). The sample was loaded into the reaction cell and a background spectrum was collected. Next, the sample was exposed to 20% CO in Ar at a flow rate of 30 mL/min for 30 min to reach the CO saturation coverage. Then we purged the sample with Ar to remove the gas phase CO from the cell, and a series of DRIFT spectra were collected during both the CO-adsorption and the Ar-purgation process (256 scans, 4 cm$^{-1}$).

**Electrochemical measurements**. All electrochemical measurements were carried out in a typical three-electrode electrochemical cell using graphite rod as the counter electrode and saturated Ag/AgCl as the reference electrode. The catalyst ink (5 mg catalyst + 765 μL water + 200 μL ethanol) was sonicated for 2 h to make the catalyst homogeneously dispersed, then 35 μL Nafion solution (5 wt%) was added to the ink and sonicated for 30 min. The as-prepared ink (10 μL) was loaded onto the glass carbon electrode to form the working electrode. Commercial 20% Pt/ C catalyst was from Shanghai River Sen Electric Co. The sample loading was determined to be ~0.25 mg cm$^{-2}$. The HER activity measurement was carried out in high purity H$_2$-saturated 0.5 M H$_2$SO$_4$ (pH 0.1) and 1.0 M KOH (pH 13.8), using linear sweep voltammetry with a scan rate of 5 mV s$^{-1}$. The HER stability was evaluated by measuring amperometric i–t curve under the overpotential ($J$ = 10 mA cm$^{-2}$). The potentials were calibrated to RHE using Pt mesh as working electrode (Supplementary Fig. 22):

$$E\,(\text{RHE, 0.5 M H}_2\text{SO}_4) = E\,(\text{Ag/AgCl}) + 0.204\,\text{V}$$

$$E\,(\text{RHE, 1.0M KOH}) = E\,(\text{Ag/AgCl}) + 1.012\,\text{V}$$

**Operando XAFS measurements**. The Pt L$_3$-edge (11,564 eV) XAFS spectra were measured at the 1W1B beamline of Beijing Synchrotron Radiation Facility (BSRF), China. The storage ring of BSRF was operated at 2.5 GeV with a maximum electron current of 250 mA. The hard X-ray was monochromatized with a Si (111) doublecrystal monochromator and detuned by 30% to remove harmonics. Operando XAFS measurements were performed in the fluorescence mode. Position of the absorption edge ($E_0$) was calibrated using Pt foil.

**XAFS data analysis**. The acquired EXAFS data were processed according to standard procedures using the ATHENA module implemented in the IFEFFIT software packages[40]. The $k^2$-weighted χ (k) data in the $k$-space ranging from 3.0 to 11.0 Å$^{-1}$ were Fourier-transformed to real ($R$) space using hanning windows (dk = 1.0 Å$^{-1}$) to separate the EXAFS contributions from different coordination shells. To obtain the detailed structural parameters around the Pt atom in the as-prepared samples, quantitative curve fittings were carried out for the Fourier-transformed $k^2$χ(k) in R-space using the ARTEMIS module of IFEFFIT. Effective backscattering amplitudes F(k) and phase shifts Φ(k) of all fitting paths were calculated with the ab initio code FEFF8.0. For the Pt$_1$/N-C samples, a $k$-range of 3.0 ~ 11.0 Å$^{-1}$ was used and curve fittings were carried out in R-space within an R

range of (1.0, 2.1) Å for $k^2$-weighted χ(k) functions. The number of independent points is given by

$$N_{ipt} = 2\Delta k \times \Delta R/\pi = 2 \times (11.0 - 3.0) \times (2.1 - 1.0)/\pi = 6$$

As for the sample under various conditions, the Fourier-transformed curves all showed a single prominent coordination peak at ~1.6 Å assigned to the Pt-C/N coordination. For the ex situ Pt$_1$/N-C sample, two separate Pt-C and Pt-N scattering paths were included for fitting. The combination of oxidation state and EXAFS fittings demonstrate that four-coordinate Pt with three Pt-C bonds and one Pt-N bond, is most reasonable for ex situ sample, as well as the open circuit and +0.5 V sample. The FT peak intensity of the sample under +0.15 V declines by 13%, which may arise from the Pt-C/N bonds breaking, together with an additional Pt-O coordination from H$_2$O adsorption. Thus, the scattering paths of Pt-C/N and Pt-O were considered. Although under −0.07 V, the FT peak intensity further declines by 35%, which is ascribed to the dissociation of the adsorbed H$_2$O followed by H adsorption on Pt, leading to only Pt-C/N path considered for EXAFS fitting. It should be mentioned that under operating conditions, the structures derived from EXAFS fitting may correspond to the species that are most present in the rate-limiting step. Hence, under +0.15 V, the water absorbed on Pt$_1$-C$_2$ or Pt$_1$-C$_1$N$_1$ structures (H$_2$O-Pt$_1$-C$_2$, H$_2$O-Pt$_1$-C$_1$N$_1$) are considered because the reaction is limited by dissociation of the absorbed H$_2$O. Although under −0.07 V, the rate-limiting step is Heyrovsky step determined by the Tafel slope, leading to high-coverage absorbed H keep intact on Pt from desorption, and thus H-Pt$_1$-C$_2$ and H-Pt$_1$-C$_1$N$_1$ structures are considered.

During curve fittings, the Debye–Waller factors ($\sigma^2$) and bond length ($R$) were treated adjustably for all samples in all paths. For the ex situ sample, $N$ and $R$ were treated as adjustable parameters for both Pt-N and Pt-C paths, whereas their $\sigma^2$ and $\Delta E_0$ were set equal in order to reduce the number of fitting parameters. Thus, the number of the adjustable parameters for the ex situ sample is $N_{para} = 6 \leq N_{ipt}$. For the +0.50 V sample, $\Delta E_0$ of the Pt-C/N path was set equal to that of the ex situ sample, other parameters were treated adjustably. $N_{para} = 6$. For the +0.15 V and −0.07 V sample, $\Delta E_0$ of the Pt-C/N path were set equal to that of the ex situ sample. In addition, the fitting strategy of the 0.15 V-pH1 sample was identical to −0.07 V sample; while the fitting strategy of +0.15 V-pH14 sample was identical to that of the +0.15 V sample. Using these strategies, all generated $R$ factors are smaller than 0.01, indicating excellent quality.

**Computational methods**. All theoretical calculations were performed using Vienna ab initio simulation packages[41]. The exchange–correlation interaction was described by the generalized gradient approximation with the Perdew–Burke–Ernzerhof functional[42]. A two-dimensional carbon/nitrogen matrix supercell was built on a 8 × 8 unit cell for the four-coordinate Pt$_1$-C$_3$N$_1$ system, to simulate the two-coordinate Pt intermediate, we modulated the pristine graphene surface with 16 (Pt$_1$-C$_2$-1, Pt$_1$-C$_1$N$_1$-1) and 15 (Pt$_1$-C$_2$-2, Pt$_1$-C$_1$N$_1$-2) carbon/ nitrogen atoms removed. The effects of the varying hydrogenation degrees of the dangling carbon atoms around the defects have been investigated (Supplementary Fig. 23). Considering that these dangling carbon atoms would undergo hydrogenation during electrochemical reduction, thus they were saturated with H atoms in the main text. The plane wave cutoff was set to 500 eV, with the convergence of energy and force set to 1 × 10$^{-5}$ and 0.01 eV Å$^{-1}$, respectively. To avoid the interaction between two adjacent layers the vacuum thickness was set to 15 Å. The Brillouin zone was sampled by a 5 × 5 × 1 $k$-point grid with the Monkhorst–Pack scheme for structural optimization and a 10 × 10 × 1 $k$-point grid for electronic structure calculations for all systems. The solvent effects were considered by using the Poisson–Boltzmann implicit solvent model with the dielectric constant set to be 78.4 for solvent water[43].

## Data availability

The data supporting the findings of this study are available from the corresponding author upon reasonable request.

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

## Acknowledgements

This research was supported by China Ministry of Science and Technology under Contract of 2017YFA0208300 and 2017YFA0402800, the National Natural Science Foundation of China (grants no. 21471143, 21533007, 11621063) and the Fundamental Research Funds for the Central Universities (KY2310000020, WK2340000076, WK2060030029), and Youth Innovation Promotion Association CAS (CX2310000091). We thank NSRL, BSRF, and SSRF for the synchrotron beam time.

## Author contributions

T.Y. developed the idea and designed experiments. S.F., X.K.L., W.L., D.H.W., W.Z., and S.Q.W. performed the catalyst synthesis and characterizations, XAFS measurement, and electrochemical experiments, collected, and analyzed the data. S.F., J.G., and J.L.L. performed ALD synthesis. X.R.Z. and Y.F.L. conducted and discussed the theoretical calculations. S.F. and Y.L. performed the aberration-corrected STEM characterization. S.F. and T.Y. co-wrote the paper. All authors discussed the results and commented on the manuscript.

## Competing interests

The authors declare no competing interests.
