## [Peer Review File · Nature Communications]

Reviewers' comments:

Reviewer #1 (Remarks to the Author):

In my original review I asked the authors to better justify they employed model for their DFT calculations. In particular, I wanted to know why the authors chose a model in which all carbon dangling bonds were saturated by hydrogen. The authors responded by saying that "the defect would change the pristine electronic property, such as conductivity and magnetism, which are depended on the edge structures of defects" and that for this reason they chose to saturate all carbon atoms. I agree that the defects will drastically impact the electronic properties of the support and that is why it is essential to explicitly consider these defects in the model. These defects are almost certainly present in the actual system and the authors need to carefully consider the effects of such defects in their analysis. I originally asked that the authors provide direct evidence to support the claim that all dangling carbon bonds would be saturated, which is still missing in the revised manuscript.

I agree with the comment from Reviewer 1, who stated that the employed models represent just one possible configuration and that a more careful justification of the local environment surrounding the Pt atom is necessary before this work is publishable.

Reviewer #2 (Remarks to the Author):

Although the authors conducted more DFT calculations to support the FT analysis, the questions on the structure of Pt single atoms and the intermediates of SAC Pt in HER still exist. Questions 1, 3, 4 were not addressed properly. The reviewer cannot support the publication of this work in Nature Communications.

1. As a monolayer of oxygen containing functional groups covered on the N-C surface, at low temperature ALD process, why did Pt atom not coordinate with O atoms, but instead coordinate to C?
2. Through comparison of adsorption energy of Pt1-C1N1 and Pt1-C2, due to larger $|\Delta G_H^*|$ of Pt1-C1N1 sites than that of Pt1-C2, the authors considered the active site is in a Pt1-C2 configuration, this description could not answer question 3.
3. Moreover, by the comparison of the adsorption energy of *H , which was calculated by DFT, to judge the configuration of intermediate state is unreasonable, and no any experimental data can support this conclusion.

Reviewer #1 (Remarks to the Author):

In my original review I asked the authors to better justify they employed model for their DFT calculations. In particular, I wanted to know why the authors chose a model in which all carbon dangling bonds were saturated by hydrogen. The authors responded by saying that “the defect would change the pristine electronic property, such as conductivity and magnetism, which are depended on the edge structures of defects” and that for this reason they chose to saturate all carbon atoms. I agree that the defects will drastically impact the electronic properties of the support and that is why it is essential to explicitly consider these defects in the model. These defects are almost certainly present in the actual system and the authors need to carefully consider the effects of such defects in their analysis. I originally asked that the authors provide direct evidence to support the claim that all dangling carbon bonds would be saturated, which is still missing in the revised manuscript.

I agree with the comment from Reviewer 1, who stated that the employed models represent just one possible configuration and that a more careful justification of the local environment surrounding the Pt atom is necessary before this work is publishable.

Reply: We thank the reviewer for pointing out this critical question. We realize that we have misunderstood the meaning of the reviewer’s question and have missed some important information in the previous response. According to the reviewer’s suggestions, we conducted additional calculations regarding the H₂O and H adsorption energies on the models with dangling carbon bonds unsaturated by hydrogen atoms. As shown in Fig. R1, the H₂O absorption energy is significantly reduced, from 0.32 eV to -0.05 eV and 0.35 eV to -0.08 eV for Pt-C₂-1 and Pt-C₂-2, respectively. This indicates that the water absorption on the Pt sites are much easier on unhydrogenated surfaces. Meanwhile the H absorption is more moderate as ΔG_H reduced from 0.17 eV to 0.06 eV and 0.07 eV to 0.01 eV for Pt-C₂-1 and Pt-C₂-2, respectively.

Fig. R1 | (a) The DFT models of hydrogenated and unhydrogenated Pt-C₂ structures. (b) The corresponding H₂O and H adsorption energies.

From these additional calculations, it is revealed that both water and hydrogen absorption are improved when the carbon dangling bonds are unsaturated by hydrogen atoms in the defects. Then, for a purpose of a better understanding of the dynamic structure during realistic reaction and the method of how we define the structure, we investigated the underlying reason for the differences of the absorption energies between the hydrogenated and unhydrogenated models. For the unhydrogenated Pt-C₂ models, the dangling bonds of edge sp² carbon atoms around the defects may strengthen the adsorption strength of the intermediates, as demonstrated by the charge redistribution around the defects (Fig. R2). Compared to the hydrogenated systems, larger localized charge density around the Pt atom is shown in the unhydrogenated systems, resulting the more favorable H₂O activation and H adsorption. The Bader

analysis also revealed that the Pt atoms are more positively charged in unhydrogenated system than that in hydrogenated system, as shown in Supplementary Table R1.

In summary, the above results demonstrate that, regardless of the hydrogenated or unhydrogenated cases, the adsorption properties on the Pt-C₂ models (both) are unambiguously superior to those on the Pt₁-C₁N₁ and the pristine Pt₁-C₃N₁ models, confirming our previous conclusions are solid. We would like to note that these theoretical models are constructed for the purpose of simulating the dynamic intermediate two-coordinate structure, thus the theoretical models might be different from the realistic structure under *operando* conditions. Furthermore, it has been reported that, during electrochemical reduction in proton-rich electrolytes, the carbon atoms with dangling bonds in graphene could be spontaneously hydrogenated by the protons (Quezada-Renteria *et al. Carbon* **149**, 2019, 722-732; Ambrosi *et al. Chem. Rev.* **114**, 2014, 7150-7188; Hallam *et al. Electrochem. Commun.* **13**, 2011, 8-11). Therefore, we employed the fully hydrogenated models in the main text, and this is also consistent with the strategies used in the previous research (Lu *et al. Nat. Commun.* **10**, 2019, 631). To address this issue, we also illustrated in detail how we constructed these models in the Computational Methods of the main text.

Fig. R2 | Calculated charge density difference of hydrogenated Pt₁-C₂-1(a) Pt₁-C₂-2(b) and unhydrogenated Pt₁-C₂-1(c), Pt₁-C₂-2 (d). The iso-surface was set as 0.06 e/Å³, where the yellow and blue contours denote electron accumulation and deletion, respectively.

Supplementary Table R1. The calculated bader charge of Pt atom in hydrogenated and unhydrogenated systems.

Species	Hydrogenated (e)	Unhydrogenated (e)
Pt ₁ -C ₂ -1	+0.11	+0.26
Pt ₁ -C ₂ -2	+0.12	+0.24

Reviewer #2 (Remarks to the Author):

Although the authors conducted more DFT calculations to support the FT analysis, the questions on the structure of Pt single atoms and the intermediates of SAC Pt in HER still exist. Questions 1, 3, 4 were not addressed properly. The reviewer cannot support the publication of this work in Nature Communications.

1. As a monolayer of oxygen containing functional groups covered on the N-C surface, at low temperature ALD process, why did Pt atom not coordinate with O atoms, but instead coordinate to C?

Reply: We thank the reviewer for this insightful question. In the former response, we claimed that O₂ was used to activate the substrate surface, and we have realized that it was actually quite misleading. Given below is our supplementary discussion, we hope this could convincingly address the reviewer's question.

When the first half cycle of Pt ALD was carried out on the N-C support, the nucleation of the MeCpPtMe₃ precursor can be very different from the ones at the stable growth regime on oxygen covered Pt surface during Pt ALD (W.M.M. Kessels et al. *App. Phys. Lett.* **95**, 2009, 013114; A.J.M. Mackus, et al. *Chem. Mater.* **24**, 2012, 1752). In our case, we expect that MeCpPtMe₃ nucleates on the sites of coordinatively unsaturated pyridinic nitrogen atoms (Equation A), since MeCpPtMe₃ can't nucleate on defect-free C surfaces, as shown in our previous work (Yan et al. *J. Am. Chem. Soc.* **137**, 2015, 10484–10487). Indeed, dissociative adsorption of ALD precursors have been frequently observed in other systems, for instance, dissociative chemisorption of ferrocene on Pt surface in our recent work (Cao, et al. *Nature*, **565**, 2019, 631-635).

Next, in the second half cycle, O₂ is used to combust off all the ligands (Equation B), and the remaining Pt is coordinated to the N atoms and likely the neighboring C atoms to minimize the surface energy. Therefore, after one ALD cycle, Pt atoms were coordinated with N/C atoms instead of O atoms.

2. Through comparison of adsorption energy of Pt₁-C₁N₁ and Pt₁-C₂, due to larger $|\Delta GH^*|$ of Pt₁-C₁N₁ sites than that of Pt₁-C₂, the authors considered the active site is in a Pt₁-C₂ configuration, this description could not answer question 3.

Last round question 3. The bond energy of Pt-N is larger than that of Pt-C, that is why the nitrogen doped carbon can promote the stability of the Pt catalysts in ORR. In the manuscript, in HER, the authors proposed that the declined intensity of Fourier transform (FT) spectra could be attributed to the breakage of Pt-N bond, and the active site transferred from Pt-3C₁N to Pt-2C, why?

3. Moreover, by the comparison of the adsorption energy of *H, which was calculated by DFT, to judge the configuration of intermediate state is unreasonable, and no any experimental data can support this conclusion.

Reply: We sincerely thank the reviewer for pointing these insightful criticisms and questions. These questions are very important and greatly help us reconsider our verifications on the intermediate structure. For better understanding, we present the reviewer's question #3 in the last round here, together with the question 2 & 3 in this round. We agree with the reviewer's opinions that the identification of the intermediate structure lacks direct experimental evidence, which is quite necessary to draw a clear conclusion on the dynamic structure from possible configurations, i.e. Pt₁-C₂ and Pt₁-C₁N₁. We are sorry to have made a misleading response on this issue in the last round, when we lack strong supports from both theoretical and experimental data. Here, we have tried our best to conduct more theoretical calculations and experimental measurements to make a comprehensive understanding on the intermediate structure. We would like to discuss the intermediate structure further, from the following aspects.

1. We calculated the bond energies of Pt-C and Pt-N, based on Pt₁-C₄ and Pt₁-N₄ configurations, shown in Fig. R3. The average bond energies of Pt-C and Pt-N are 0.114 eV and 0.365 eV, respectively. It is true that Pt-N bond is stronger than Pt-C bond, thus we agree with the reviewer that Pt₁-N₄ is more stable than Pt₁-C₄. As such, Pt₁-C₁N₁ might also be more stable than Pt₁-C₂ if they are the initial structure; but the cases are complicated during the reaction. In the case we study, there are

two possible pathways for the hydrogen evolution reaction on the Pt₁-C₃N₁ surface, simply noted as:

(1) The Pt₁-C₂ pathway:

(2) The Pt₁-C₁N₁ pathway:

*Note that the *OH and *H species desorbed from Pt site are omitted.*

Based on the illustration above, the reaction pathway via which the reaction would proceed is not correlating much with the Pt-C and Pt-N bond energies, it is determined by the energy barrier of the reaction instead. From Fig. 3b in the main text, we notice that the ΔG of the hydrogen desorption on the Pt₁-C₁N₁ (H-Pt₁-C₁N₁ → Pt₁-C₁N₁) is 0.9 or 1.35 eV, much larger than that of the Pt₁-C₂ (0.17 or 0.06 eV). This large ΔG could possibly represent a very large energy barrier for the Pt₁-C₁N₁ pathway. Therefore, both Pt₁-C₂ and Pt₁-C₁N₁ are possible intermediate configurations, but we agree with the reviewer that it is not reasonable to directly determine the configuration of the intermediate state by comparing the H adsorption energies, thus, we consider that the Pt₁-C₂ is likely but not exclusively to be the intermediate structure.

Fig. R3 | The Pt₁-C₄ (left) and Pt₁-N₄ (right) models. The average bond energies of Pt-C and Pt-N are 0.114 and 0.365 eV, respectively.

2. To try our best to determine the intermediate structure, besides the theoretical investigation, we also conducted wavelet transform (WT) analysis based on the XAFS data, by which we hoped to find some differences between Pt-C, Pt-N and Pt-O coordination. The WT contour spectra were shown in Fig. R4.

For the PtO₂ sample, the WT intensity maximum near 3.73 Å⁻¹ can be attributed to the Pt-O coordination. For the ex-situ sample, the WT intensity maximum locates at ~ 3.55 Å⁻¹, a little lower K than that of PtO₂, indicating that Pt coordinates with lighter atoms than O, i.e. C and/or N atoms. For the +0.50 V sample, the position of the WT center remain unchanged, consistent with the previous EXAFS analysis in the main text. Interestingly, the WT center of + 0.15 V sample shift to higher K, close to that of PtO₂, implying an extra Pt-O coordination under +0.15 V, likely arising from H₂O adsorption on Pt sites. Therefore, the WT results are highly consistent with the previous EXAFS fitting results.

In summary, the Pt single atom locates at a multi-coordination environment with C, N and O atoms, of which the scattering amplitude are very close. The above WT analysis provides evidence for the coordinative evolution of Pt, such as the intermediate oxo group adsorption. While for the discrimination of Pt-C and Pt-N coordination, as the lack of XAFS data of pure Pt-C_x and Pt-N_x standards, it is still rather difficult to identify the intermediate configuration from Pt₁-C₂ and Pt₁-C₁N₁.

Fig. R4 | Wavelet transform for the k^2 -weighted EXAFS data of different conditions.

3. Finally, we resort to *operando* Fourier transform infrared (FTIR) and Raman tests, hopefully to acquire as much information about the dynamic evolution of Pt sites as we can. The FTIR and Raman spectra are plotted in Fig. R5. Unfortunately, but not surprisingly, no novel signal was detected under wide-range potentials, both in FTIR and Raman tests. It is reasonable that the IR signal of hydrogen evolution intermediates is highly interfered by the strong water absorption. Besides, the much lower Pt content of our single-atom catalysts than that of the bulk materials, makes it harder for *operando* tests, though the single-atom Pt has superior HER activity.

On the basis of the above results, it is difficult to directly distinguish whether the intermediate structural configuration is $\text{Pt}_1\text{-C}_2$ or $\text{Pt}_1\text{-C}_1\text{N}_1$, taking into consideration of the currently available experimental techniques.

Fig. R5 | Operando FTIR (a) and Raman (b) spectra. All potentials are relative to RHE.

With all the theoretical and experimental efforts, we consider that we still have made advances in understanding of the dynamic structure of single atomic Pt site, both in the electronic and atomic point of view. Here, we would like to highlight again the importance of XAFS data analysis (XANES analysis, EXAFS fitting, wavelet transform, etc.), and its combination with DFT calculation to identify the dynamic structure of single Pt site, and to reveal reaction mechanism for deeply understanding the “structure-activity relationship”:

- i. We have scrupulously determined the ex-situ configuration of our sample, i.e. Pt₁-C₃N₁, and verified it by combining EXAFS curve fitting with oxidation states fitting from both the Δ XANES data and the Bader charge simulation.
- ii. Based on EXAFS fitting, XANES analysis, and WT calculations, we can conclude that the single atomic Pt sites were partially released from the substrate, with the atomic and electronic structural structures evolving into the two-fold Pt-C/N coordination and the near-free state, respectively.
- iii. By DFT calculation, we reveal that the two-coordinate Pt intermediate exhibits optimized H₂O and H adsorption energies, which lead to the superior HER activity

in wide-pH range. Further, we investigated the Pt-H interaction by analyzing the H 1s-Pt 5d hybridization based on PDOS (added in this version), which unravel the underlying reason for the favorable hydrogen adsorption of the two-coordinate Pt intermediate.

Despite these innovative discoveries, we acknowledge that one problem still remains unsolved to determine the atomic configuration of the intermediate two-coordinate Pt from Pt₁-C₂ and Pt₁-C₁N₁. We also realized that, with the state-of-the-art techniques, it might be unlikely to distinguish Pt₁-C₂ and Pt₁-C₁N₁ from each other in such heterogeneous catalyst under reaction conditions.

Based on this fact, we agree with the reviewer that it is unreasonable to exclusively identify Pt₁-C₂ as the two-coordinate intermediate by XAFS technique without other direct experimental evidence. Accordingly, we seriously modified the statements in our manuscript, from the following points:

- i. In the EXAFS fitting, we supplemented the possible two-coordinate configuration of Pt₁-C₁N₁, proposing that both Pt₁-C₂ and Pt₁-C₁N₁ are possible intermediate configurations.
- ii. In the energetic analysis in Fig. 3, we presented the adsorption energies of both Pt₁-C₂ and Pt₁-C₁N₁ models for clear comparison, without speculative conclusion of the specific Pt₁-C₂ intermediate configuration. The statements using Pt₁-C₂ as the intermediate have been altered to “two-coordinate sites” and “Pt₁-C₂ and/or Pt₁-C₁N₁”. Furthermore, we clarify that some calculations, such as PDOS, did not involve the Pt₁-C₁N₁ models, to solely focus on the superior energetic performance of the Pt₁-C₂ models.
- iii. We added more PDOS analysis (Fig. 3c and 3d), which provided more insights about the near-free state of the single Pt sites, and further revealed the underlying mechanism of the stronger Pt-H interaction for the Pt₁-C₂ sites.

Finally, we would thank the reviewer again for providing these criticisms, which greatly help us to improve the quality of this manuscript. After the above modifications,

the main idea of this paper, which focus on the dynamic near-free state of single Pt site and the underlying mechanism for the superior wide-pH-range HER performance, is more clear and strict. We believe this work provides a new concept in single-atom catalysis, as well as in design and synthesis of high-performance single-atom catalysts. Moreover, the relevant ideas have always been research hotspots. For example, to tune the electronic structure of the active sites by means of metal alloying, the metal-adsorbate interaction is optimized and then catalytic property is improved (Stamenkovic *et al.* *Angew. Chem. Int. Ed. Engl.* **45**, 2006, 2897-2901; Studt *et al.* *Science* **320**, 2008, 1320–1322; Greeley *et al.* *Nat. Chem.* **1**, 2009, 552–556; Greiner *et al.* *Nat. Chem.* **10**, 2018, 1008-1015). Combing these ideas together, we are able to depict a more complete map for guiding rational catalysts design and synthesis.

REVIEWERS' COMMENTS:

Reviewer #1 (Remarks to the Author):

In the revised manuscript the authors have conducted additional DFT calculations to more carefully consider the effect that varying degrees of hydrogenation at the active site has on the computed adsorption energies. I encourage the authors to report this data in the supplementary material (currently it only appears in the response letter). These calculations answered my previous comment and this work is now acceptable for publication.

Reviewer #2 (Remarks to the Author):

The authors have made significant changes to the manuscript. Although not all questions raised are properly answered, the reviewer believe the current version can be considered for publication.

Reply to the reviewers' comments:

We are sincerely grateful to all the reviewers for their positive advices for the publication of our manuscript. Below we provide point-by-point replies to the Reviewers' comments.

Reviewer #1 (Remarks to the Author):

In the revised manuscript the authors have conducted additional DFT calculations to more carefully consider the effect that varying degrees of hydrogenation at the active site has on the computed adsorption energies. I encourage the authors to report this data in the supplementary material (currently it only appears in the response letter). These calculations answered my previous comment and this work is now acceptable for publication.

Reply: We appreciate the reviewer for the positive suggestions and the recommendation for the publication of our work.

According to the reviewer's suggestions, we have added the results that include adsorption energies of both hydrogenated and unhydrogenated Pt₁-C₂ models (as shown in the added Supplementary figure 23), and added the necessary statements in Computational Methods in the main text.

Reviewer #2 (Remarks to the Author):

The authors have made significant changes to the manuscript. Although not all questions raised are properly answered, the reviewer believe the current version can be considered for publication.

Reply: We are grateful to the reviewer for these nice comments and strong support on the publication of our work.